# Time-Series Comparative Transcriptome Analyses of Two Potato Cultivars with Different Verticillium Wilt Resistance

**DOI:** 10.3390/plants15010026

**Published:** 2025-12-21

**Authors:** Guoquan Fan, Zhiguo Ren, Yanling Gao, Guili Di, Peng Wang, Shu Zhang, Wei Zhang, Min Tu, Yin Li, Shuxin Han

**Affiliations:** 1Institute of Industrial Crops, Heilongjiang Academy of Agricultural Sciences, Harbin 150086, China; fanguoquan@haas.cn (G.F.); diguili@haas.cn (G.D.); wangpengmls@163.com (P.W.); zhangshu@haas.cn (S.Z.);; 2Heilongjiang Provincial Key Laboratory for Potato Biology and Quality Improvement, Harbin 150086, China; 3College of Plant Protection, Northeast Agricultural University, Harbin 150030, China; b240301004@neau.edu.cn; 4Hubei Key Laboratory of Agricultural Waste Resource Utilization, School of Chemical and Environmental Engineering, Wuhan Polytechnic University, Wuhan 430023, China; 5The International Science and Technology Cooperation Base (Genetic Engineering), The Key Laboratory of Molecular Biophysics of Chinese Ministry of Education, College of Life Science and Technology, Huazhong University of Science and Technology, Wuhan 430074, China; yinli2021@hust.edu.cn

**Keywords:** comparative transcriptome, potato, verticillium dahliae, transcriptomics, phenylalanine ammonia-lyase

## Abstract

Comparative omics analysis offers one of the most direct and effective approaches to gain novel insights into crop traits, facilitating candidate gene identification and crop improvement. *Verticillium dahliae* causes one of the most globally devastating crop diseases, the Verticillium wilt (VW). However, comparative transcriptome resources regarding VW resistance remain scarce in a major host species potato. To address this knowledge gap, we provide a comprehensive comparative RNA-seq analysis of VW resistance between a VW-resistant and -susceptible potato cultivar (LS8 and SP, respectively). VW inoculation induced dramatic transcriptomic changes, resulting in 14,310 differentially expressed genes (DEGs) in LS8 and 21,739 DEGs in SP. With the time-series analysis, we disentangled the VW-associated transcriptomic responses from those reflected LS8-SP cultivar differences. Particularly, LS8 featured a rapid response of phytohormone salicylic acid and defense-related specialized metabolites at 1 day post inoculation (dpi), followed by large-scale metabolic reprogramming, including carbohydrate and choline metabolism and enhanced biosynthesis of secondary cell wall components (e.g., hemicellulose, xylan, cuticle, suberin, and wax). Furthermore, we identified highly expressed VW-responsive potato phenylalanine ammonia-lyase genes (*StPAL*s) and revealed the higher PAL activities in LS8 associated with VW resistance. Overall, our results provide the first transcriptomic insights into VW resistance in potato and new candidate genes regarding VW resistance.

## 1. Introduction

Potato (*Solanum tuberosum* L.), a dicotyledonous plant of the Solanaceae family, is the world’s largest non-cereal crop, grown over approximately 17.8 million hectares, and serves as a staple food for over one billion people [1,2]. Verticillium wilt (VW), mainly caused by *Verticillium dahliae* (*V. dahliae*), is a major crop disease that occurs in agriculture and affects over 200 plant species, including potato, cotton, and tomato, causing yield reductions of 10–60% and significant economic losses [3,4]. Yield losses of potato due to Verticillium wilt range from 10% to 50% [5,6,7]. *V. dahliae* infects its host crops from the roots and underground stem tissues and forms hyphae in the vascular tissue, with the hyphae growing along the vascular tissue to the above-ground stems and leaves to cause symptoms of chlorosis from the bottom leaves to the upper leaves.

China is the largest potato producer in the world, and VW has a high incidence in China’s potato-growing areas, causing huge economic losses. Moreover, *V. dahliae* persists and accumulates in the soil in the form of microsclerotia, increasing the incidence of VW through continuous monocropping [8]. Although crop rotation can reduce the impact of soil-borne diseases on crop plants, it is not sufficiently effective for *V. dahliae*, which has a long latency period in the soil [4]. The persistent presence of *V. dahliae* in the soil also makes it difficult to avoid recurrent plant infections. Reducing Verticillium propagules in the soil through chemical soil fumigation is expensive and adversely affects beneficial soil microbial communities, human health, and the environment [9,10]. Utilizing potato varieties that are VW-resistant represents the most effective strategy for VW disease management.

Confronted with diverse environments, plants exhibit remarkable adaptability, and their intrinsic genetic regulatory mechanisms offer both promise and challenges for the improvement of quantitative traits, particularly in plant disease resistance. Elucidating the genetic regulatory mechanisms of varieties harboring disease resistance genes is crucial for crop improvement and breeding. Potato is an autotetraploid and highly heterozygous species, implying that each cell contains four sets of chromosomes with a substantial amount of genetic variation among them. This complex genomic structure complicates genetic analysis and the localization of resistance genes. The narrow genetic base of most existing commercial potato cultivars further increases the difficulty of studying resistance traits. In 2004, breeding scientists tracked the origin of susceptibility and resistance to Verticillium wilt in 39 North American potato cultivars through molecular marker analysis. Due to the methodological limitations, only the STM1051–193 allele was identified as potentially associated with VW resistance [11]. After the advent of sequencing technologies, new challenges emerged in potato genome sequencing. Since the potato genome is autotetraploid, early sequencing technologies produced short reads, making genome assembly highly challenging. Recently, the application of new technologies such as single-molecule sequencing has enabled researchers to obtain longer and more accurate reads, thereby improving the quality of genome assembly. The development of single-cell sequencing technology allowed researchers to more effectively perform haplotyping, resulting in the successful assembly of the complete genomes of tetraploid potato cultivars [12,13,14,15]. In 2025, construction of the potato pan-genome further addressed the complexity of the potato genome and the diversity across potato accessions and relative species [16]. These achievements lay genomic foundation for understanding molecular mechanisms of VW resistance in potato germplasm.

Over the past years, extensive research has been conducted to investigate the genetic basis and to develop breeding strategies for enhancing crop resistance to VW. Compared to the related research in potato, those in cotton, another major host plant species of VW, reached fruitful results. In cotton, genome-wide association studies (GWAS) and transcriptome-wide association studies (TWAS) studies using multiple transcriptomic datasets and diverse germplasm resources revealed the genetic and regulatory basis of VW resistance [17]. Key findings include the interaction of Vd06254 with GhMYC3, a jasmonic acid-pathway transcription factor, inhibiting strigolactone synthesis via GhCCD8 to disrupt resistance [18]. GhAMT2 is another VW resistance regulator linked to lignin biosynthesis, salicylic acid signaling, and ROS scavenging modules [19]. In potato, GWAS analysis for 162 tetraploid potato materials identified genetic loci associated with VW resistance, highlighting the loci on chromosome 5 that is co-located with the maturity gene StCDF1. This result implicates a potential genetic link between tuber maturation timing and resistance to this pathogen [20]. Another genetic study in potato revealed that miR482e negatively regulates VW susceptibility, while the target genes encoding the nucleotide binding site (NBS) and leucine-rich repeat (LRR) proteins are positively contributing to VW resistance. MiR482e also triggered the production of trans-acting (ta) -siRNA targeting defense-related proteins to control VW susceptibility, indicating the involvement of defense-related genes in potato VW resistance [21]. More recently, studies on the VW pathogen reported the genome of *V*. *nonalfalfae* and provided new insights into the pathogenic mechanisms of VW, highlighting 21 differentially expressed genes (DEGs) in *V. nonalfalfae* during VW infection which encode proteins with low-complexity regions (LCRs) and transmembrane domains [22]. In parallel, recent efforts using the green fluorescence protein (GFP)-expressing *V. dahliae* isolate VD01 helped elucidate the infection dynamics of *V. dahliae* in potato plants: within 24 h post inoculation, *V. dahliae* spores began to germinate; after 7 days, the hyphae expanded from the roots to the cortex but not to the stem vascular tissues; after 14 days, hyphae largely formed in the vascular tissues and expanded fully to the stems within 21 days post inoculation [23,24].

As large-scale gene identification related to VW resistance and mechanistic studies with molecular genetic approaches remain scarce in potato, we propose here that comparative transcriptomic analysis offers one of the most direct and effective approaches to gain novel insights into crop traits, in this case VW resistance. In the present study, we selected two cultivars, Shepody (SP) and Longshu 8 (LS8), as VW-susceptible and -resistant, respectively. We performed VW inoculation and mock inoculation (as the control) to obtain time-series samples for comparative RNA-seq analyses. The major aim of our study was to provide a comparative RNA-seq dataset of VW resistance in potato and to unravel the complexed dynamic responses of gene expression upon VW inoculation. With time-series transcriptomics analyses, we offer new molecular insights into the dynamic responses of several metabolic pathways to VW inoculation in LS8 tissues. We demonstrated that the enhanced biosynthesis of several secondary cell wall (SCW) components (e.g., hemicellulose, xylan, cuticle, suberin, and wax) could contribute to better VW resistance. We identified several potato phenylalanine ammonia-lyase genes (*StPAL*s) as the candidate for the enhanced SCW metabolism.

## 2. Results and Discussion

### 2.1. The Two Potato Cultivars Showed Contrasting Resistance to Verticillium Wilt

To obtain molecular insights into Verticillium wilt (VW) resistance of potato cultivars, we first selected two cultivars (Shepody (SP) and Longshu 8 (LS8), with SP being susceptible to VW and LS8 resistant to VW). The seedling plants of the two cultivars were inoculated with VW (*Verticillium dahliae*), respectively, with the mock inoculation used as the control (regarding the details of *V. dahliae* strain and mock, see “Section 3.1 and Section 3.2”). The tear-bottom inoculation method was employed, and the main root and its adjacent underground stem tissues were used for inoculation and for sampling for RNA-seq and phenylalanine ammonia-lyase (PAL) enzymatic assay. After 30-day inoculation, SP exhibited many yellow leaves, and the stem of the plants became thinner, while LS8 showed no obvious difference for the plant morphology (Figure 1A). We isolated the inoculated tissue and visualized longitudinally. For the VW-susceptible cultivar SP, the VW-inoculated part of underground stem became browner than that of the mock-inoculated plants, showing the typical *V. dahliae* infection symptom on the vascular tissue of potato plants; by contrast, the VW-inoculated part of underground stem from VW-resistant cultivar LS8 did not show difference compared with that of the mock inoculation (Figure 1B).

To capture the dynamic changes in gene expression over the VW inoculation process and to reveal the gene expression differences associated with LS8’s VW resistance, we set up the comparative study as follows: **(1)** For both cultivars, healthy plants at 4–6 leaf stage were used for the inoculation of mock or *V. dahliae* strain with three biological replicates; **(2)** within each replicate, twenty-seven plants per cultivar were used, and three plants were sampled at each sampling stage for the mock or *V. dahliae* inoculation; the tissues collected from three plants were pooled, with one aliquot for RNA-seq and another aliquot for PAL enzymatic assay; **(3)** there are nine samples for each cultivar, with the above samples designated as follows: using the cultivar SP as an example, “SP_0” means the sample before inoculation, and “SP_1C”, “SP_3C”, “SP_5C”, and “SP_13C” represent the mock-inoculated samples 1 day, 3 days, 5 days, and 13 days post inoculation, respectively. “SP_1T”, “SP_3T”, “SP_5T”, and “SP_13T” represent the *V. dahliae*-inoculated samples 1 day, 3 days, 5 days, and 13 days post inoculation, respectively (experimental design diagram in Figure 1C upper panel; sample names shown in Figure 1D); **(4)** we specifically selected the above stages to capture the dynamic changes of gene expression during *V. dahliae* colonization and propagation in the vascular tissue and its expansion into the stem (see “Section 3.1”).

With the RNA-seq analysis, we obtained 54 RNA-seq samples (2 cultivars, 9 samples, and 3 replicates) with high sequencing quality. After quality control, these samples contained ~35 to ~67 million clean reads per sample (~5.3 Gb to ~10.0 Gb clean bases per sample) (Appendix A). The averaged Q20 sequence percentage per sample reached 98.94%, and Q30 percentage reached 95.52%. These RNA-seq data were mapped to the haploid-resolved genome of a cultivated tetraploid potato Cooperation-88 (C88) [12], reaching an averaged mapping rate of 92.6%. The uniquely mapped rate per sample ranges from 45.5% to 51.3% with an average rate of 48.2%, while the multi-mapped reads account for 48.7% to 54.5% of the total clean reads, with an averaged multi-mapped rate of 51.8% (Appendix A). This phenomenon of high-level multi-mapped rates match well with the polyploid nature of the potato genome. We detected a total of 71,739 expressed genes (48.2% of the annotated gene models) across all the samples (at least five reads per replicate sample and averaged FPKM > 0.5 in at least one dpi of one cultivar) (Appendix A). Based on the expressed genes, the result of principal component analysis (PCA) highlighted that LS8 and SP samples were well separated (Figure 2). In agreement with the PCA result, the replicated RNA-seq samples were highly correlated, among the 54 RNA-seq samples over 95% of the pairwise comparison between the replicated samples having a Pearson correlation coefficient > 0.8 (Appendix A). For the PCA result, the first and second principal components (PCs) were associated with both cultivar and VW inoculation, with the first PC and second PC accounting for 71.22% and 9.16% of expression variance, respectively. The transcriptomes of SP changed dramatically after VW inoculation, and the mock and VW-inoculated samples per dpi could be separated (Figure 2). By contrast, the transcriptomes of LS8 were not well separated, with only LS8_13C (mock) and LS8_13T (VW inoculated) clearly separated. This suggests that the LS8 transcriptomes were relatively stable after VW inoculation compared with those of SP. In addition, LS8 transcriptomes and SP transcriptomes were distinctly separated in the PCA results, suggesting the intrinsic differences in overall gene expression profiles between LS8 and SP regardless of VW inoculation (Figure 2).

### 2.2. Complexed Transcriptomic Changes Associated with VW Inoculation

To identify the transcriptomic responses associated with VW inoculation and the VW resistance of LS8, we calculated differentially expressed genes (DEGs) by pairwise comparisons between different samples (DEGs defined as log2Fold Change ≥ 1 or ≤ −1 and q value < 0.05). To fully capture the genes associated with the VW resistance of LS8, we employed two analytical approaches: **(1)** We first analyzed the DEGs of each cultivar separately and focused on identifying DEGs between mock and VW treatments at a given stage, followed by gene ontology (GO) enrichment analyses to highlight representative biological processes. Then, we compared the biological process between LS8 and SP to interpret why LS8 exhibits VW resistance. **(2)** We identified all DEGs via pairwise comparisons between different days post-inoculation (dpi) (using the LS8 cultivar as an example: LS8_0, LS8_1C, LS8_3C, LS8_5C, and LS8_13C samples or LS8_0, LS8_1T, LS8_3T, LS8_5T, and LS8_13T samples). This approach allows us to detect all DEGs across a time-series of mock or *V. dahliae* inoculation. Theoretically, this approach holds another advantage: VW-responsive genes may be differentially expressed in both cultivars and thus detected by the first approach but not necessarily linked to LS8’s VW resistance. However, identifying the genes with distinct time-series expression patterns between LS8 and SP could more effectively capture the genes contributing to LS8’s VW resistance. Using these two DEG analytical approaches, we not only compared mock versus VW treatments at each stage for each cultivar but also across different stages (Figure 3A; Appendix A), with representative DEG comparisons listed in Table 1. The 28 comparison groups in Table 1 are also indicated in Figure 3B. Accordingly, we found a total of 47,979 DEGs, which include 21,793 from SP sample comparisons, 14,310 from LS8 sample comparisons, and 38,036 DEGs from LS8 vs. SP VW-inoculated sample comparisons (Figure 3A). The large number of DEGs (38,036 genes) between LS8 and SP samples was consistent with the dramatic transcriptomic differences revealed by PCA results (Figure 2).

#### 2.2.1. VW-Responsive Genes in LS8 and SP

To detect the VW responsive genes, we compared VW-treated and mock-treated samples at the same days post inoculation (dpi) in LS8 or SP. Interestingly, the two cultivars exhibited distinct transcriptomic responses dynamically. VW-resistant LS8 generally had a higher number of DEGs between 1 and 5 dpi compared to SP, while SP had 2748 downregulated genes at 13 dpi (Table 1). In addition, we combined all VW-responsive DEGs identified at each dpi per cultivar, forming the VW-responsive DEGs in LS8 and SP, respectively. These VW-responsive DEGs should not contain those induced by wound responses which are common to both mock and VW treatments. The upregulated and downregulated genes in LS8 or SP at a given dpi were subject to gene ontology (GO) enrichment (gene count per GO term ≥ 3, *p* < 0.05, enrichment factor ≥ 1.5). We compared GO terms for upregulated or downregulated genes between LS8 and SP at each dpi to gain insights into their distinct transcriptomic responses (Figure 4).

At 1 dpi, LS8-upregulated genes were enriched in terms related to abscisic acid (ABA) response (GO:0009737), secondary metabolites (especially terpenoids) (GO:0051762, GO:0042214, GO:0046246), and responses to biotic stress, fungus, chitin, and oxidative stress (GO:0010200, GO:0009620, GO:0006979, GO:0043207). In contrast, LS8-downregulated genes were enriched in protein catabolic processes (GO:0030163), secondary cell wall (including its component xylan biosynthesis; GO:0009057, GO:0045492, GO:0009834, GO:0071669), and cytoskeleton organization (GO:0007010). This indicates that LS8 might rapidly respond to *V. dahliae* inoculation and trigger metabolic responses, such as secondary cell wall modifications and terpenoid production. SP-upregulated genes at 1 dpi were mainly involved in carbohydrate transport (GO:0015770, GO:0051119), photosynthesis and energy metabolism (GO:0019684, GO:0006091), and amino acid homeostasis (GO:0080144). SP-downregulated genes at 1 dpi were related to hormone metabolism and responses (GO:0042445, GO:009753), starch metabolism (GO:0005982), and glutamine catabolic processes (GO:0009065).

At 3 dpi, LS8-upregulated genes were enriched in organ development and starch metabolism (GO:0048366, GO:0048364, GO:0099402, GO:0048367, GO:0019252, GO:0005982), whereas LS8-downregulated genes were involved in responses to fungi (GO:0050832), external stimuli (GO:0009605, GO:0006952), and wounding (GO:0009611), suggesting that LS8 tissues maintained active organ development and primary metabolism at this stage. By contrast, SP-upregulated genes at 3 dpi were enriched in sucrose degradation, nitrogen compound transport, and hexose response (GO:00016157, GO:0071705, GO:0009746). SP-downregulated genes at 3 dpi cover diverse functions: JA response, nitrogen compound biosynthesis, and ubiquitin-dependent protein catabolism (GO:0009753, GO:0044271, GO:1901566, GO:0006511).

At 5 dpi, LS8-upregulated genes were associated with protein subunits and ribosomes, cytoskeleton organization, and cell wall biogenesis (GO:0003735, GO:0071822, GO:0007010, GO:0042546, GO:0071554). LS8-downregulated genes were primarily involved in JA biosynthesis and signaling, as well as fungus and wound responses (GO:0009867, GO:0009694, GO:0009611, GO:0009620). Comparatively, SP-upregulated genes were enriched in JA response, hormone signaling pathways, defense response, and fungus response (GO:0009753, GO:0009755, GO:0009752, GO:0009814, GO:0050832). SP-downregulated genes were related to cell cycle, nitrogen compound metabolism, and the responses to nutrient and starvation (GO:000278, GO:0007049, GO:1901564, GO:0031667, GO:0042594).

At the last transcriptome stage (13 dpi), LS8-upregulated genes were enriched in terpenoid metabolism and the responses to defense, SA, and biotic stress (GO:0080016, GO:0045338, GO:0006952, GO:0009751, GO:0009620, GO:0043207). SP-upregulated genes were mainly involved in hormone stimulus response, the regulation of cell communication and signal transduction (GO:0032870, GO: GO:0010646, GO:0009966), and polysaccharide biosynthesis (GO:0000271, GO:0044264, GO:0034645). SP-downregulated genes were associated with organ development, chlorophyll metabolism, terpenoid metabolism, and glutamate catabolism (GO:0048366, GO:0022622, GO:0099402, GO:0015994, GO:0006721, GO:0006538; Figure 4).

Overall, differences in enriched GO terms between LS8 and SP indicated that LS8 mounted a rapid response to VW inoculation and continued organ development, whereas the responses of defense and to fungus were suppressed in SP prior to 5 dpi. Carbohydrate metabolism and sugar transport were upregulated within 13 dpi, potentially supplying soluble sugars to support *V. dahliae* growth. In addition, the metabolism, signaling, and response to phytohormones (especially JA and SA) were enriched in upregulated or downregulated genes in LS8 and SP, suggesting these pathways are important for the dynamic response to *V. dahliae* inoculation.

#### 2.2.2. Identifying Genes Clusters with Distinct Expression Patterns Between LS8 and SP

We reasoned that, at the transcriptome level, the genes associated with VW resistance in LS8 should exhibit differences in dynamic expression patterns between LS8 and SP during the time course of VW inoculation. To identify genes with different dynamic expression patterns between these two cultivars, we established a stepwise RNA-seq analysis workflow (the aforementioned second RNA-seq analytical approach) to cluster genes sharing similar expression patterns across LS8 and SP samples (Figure 3A). First, we compared gene expression at different stages across LS8 and SP samples, respectively, and identified 14,310 LS8-DEGs and 21,793 SP-DEGs (Figure 3A). Collectively, these 21,793 SP-DEGs and 14,310 LS8-DEGs together accounted for 29,729 DEGs, with an overlap of 7936 DEGs between the two sets. Secondly, we employed, Mfuzz, a noise-robust clustering method, to separately group these 14,310 LS8-DEGs and 21,793 SP-DEGs into distinct gene clusters, respectively [25]. The Mfuzz analysis results exhibited that the LS8-DEGs were initially clustered into 40 LS8 gene clusters (Appendix A), while the SP-DEGs were divided into 45 SP clusters (Appendix A). For each cultivar, cluster eigengenes, representative values for each cluster, were calculated using the first principal component of the cluster. These eigengenes were then used to merge gene clusters with highly correlated dynamic expression patterns, defined by a Pearson correlation coefficient (PCC) threshold for *r* ≥ 0.78. This merging step yielded 13 and 14 gene modules in LS8 and SP, respectively (Figure 3B). Gene modules were constructed by integrating multiple original clusters to capture the core dynamic expression patterns in LS8 and SP post the mock and VW treatments. This merging step was necessary because the initial Mfuzz clustering generated an excessive number of gene clusters in both cultivars, which would have hindered downstream analyses. Visualization of these LS8 and SP gene modules revealed that, in LS8, modules M2, M6, M10, M11, M12, and M13 were primarily responsive to VW inoculation, while in SP, the VW inoculation-responsive modules were M2, M4, M8, and M14 (Appendix A).

Subsequently, we analyzed the expression profiles of all 29,729 DEGs across LS8 and SP. Given that only the aforementioned 13 LS8 modules and 14 SP modules were mainly responsive to VW inoculation, these 29,729 DEGs included 119 module combinations (covering 12,797 DEGs) where genes responded to VW inoculation in either LS8 or in SP samples. Consequently, genes within these 12,797 DEGs that displayed divergent expression profiles between LS8 and SP are hypothesized to be associated with the distinct VW resistance phenotypes of the two cultivars. To further refine this set of 12,797 DEGs, we integrated hierarchical clustering (based on gene expression patterns) and PCC values between modules, which allowed us to classify the 12,797 DEGs into 33 gene groups (Figure 5A). These groups, designated G1 to G33, each represent a unique expression pattern in response to VW inoculation (Figure 5A).

### 2.3. Functions Associated with VW-Responsive Expression Groups Underline Cell Wall Metabolism Differences Between the Cultivars

To comprehend the expression patterns of these 33 groups and their association with VW resistance in both cultivars, we visualized their expression dynamics with a heatmap. Given that SP and LS8 are VW-susceptible and VW-resistant, respectively, we reasoned that the candidate genes contributing to the differential VW resistance between LS8 and SP could be narrowed down by the following criteria: (1) The genes should exhibit VW-responsiveness in at least one cultivar, as indicated by their classification as DEGs in either LS8 or SP RNA-seq datasets; (2) the VW-responsiveness of these genes, either upregulation or downregulation, should differ between LS8 and SP (e.g., a gene responds to VW inoculation in LS8 but not in SP). Accordingly, we identified twelve gene groups (i.e., G1, G2, G5, G7, G10, G12, G13, G14, G17, G18, G19, and G20) that are VW-responsive in LS8 but display contrasting expression patterns in SP. Additionally, seven groups (i.e., G21, G22, G23, G24, G25, G26, and G27) were found to exhibit VW-responsive expression in SP while showing distinct expression profiles in LS8 (Figure 5B). For example, in G1, G10, and G19, gene expression levels were upregulated at 3 dpi following VW treatment in LS8, whereas the expression profiles were similar between the VW inoculation and mock SP samples at 3 dpi. Groups G5, G7, and G13 displayed marked upregulation at 5 dpi in VW-inoculated LS8 samples, whereas SP exhibited comparable expression levels between VW and mock treatments. Similarly, G27 showed significant VW-induced upregulation in SP at 3 dpi but was not responsive to VW in LS8 samples. Group G23 showed VW-triggered upregulation in SP at 13 dpi, while LS8 exhibited no VW responsiveness for genes in this group. Collectively, the abovementioned gene groups reflect those potentially contributing to the distinct VW resistance phenotypes between LS8 and SP.

More importantly, LS8 VW-responsive DEGs and SP VW-responsive DEGs (Appendix A) represent valuable gene sets to further highlight key gene sets deserved detailed analyses, because LS8 VW-responsive DEGs and SP VW-responsive DEGs should not contain those induced by wound responses which are common to both mock and VW treatments. To this end, the number of genes that overlapped between each gene group and LS8 VW-responsive DEGs, SP VW-responsive DEGs, or LS8/SP VW-responsive DEGs were calculated and are indicated in Figure 5B. In several gene groups, LS8 VW-responsive DEGs and SP VW-responsive DEGs comprise high percentages (more than 45% of the total number in the corresponding gene set; Appendix A), suggesting that these gene groups should enrich the DEGs identified by the aforementioned two RNA-seq analytical approaches. These gene groups include G1, G2, G5, G7, G10, G12, G13, G17, G18, G19, G20, G21, G22, and G23.

We employed gene ontology (GO) enrichment analysis to explore the representative biological functions of the aforementioned gene groups (enrichment determined via Hypergeometric test, with *q* values < 0.05 and enrich factor > 1.5; details in “Section 3.4”). Statistically significant GO terms are provided (Data S1). Among gene groups G1, G10, and G19 which encompass VW-induced genes in LS8 at 3 dpi, G1 was enriched in functions related to polysaccharide metabolism (e.g., GO:004264, GO:0005976), while G10 was enriched in GO terms associated with choline and phosphatidylcholine biosynthesis (e.g., GO:0042425, GO:0006656). Group G19 exhibited a diverse range of enriched functions, including phytohormone metabolism (GO:0034754, GO:0009690), lipid metabolism (GO:0008610), and secondary terpenoid metabolism (GO:1901928) (Figure 6). Collectively, these results suggest that the biosynthesis and metabolism of polysaccharides, certain terpenoids, choline and phosphatidylcholine, and phytohormones are VW-associatively up-regulated in LS8 at 3 dpi.

At 5 dpi, groups G5, G7, and G13 represent the genes that were highly induced in LS8 but not in SP following VW inoculation. Interestingly, G5 showed an LS8-specific upregulation pattern (with no significant change in SP) and contained genes predicted to function in the response to the phytohormone abscisic acid, biotic stimulus, and bacterium. This indicated that defense-related genes were upregulated in LS8 at 5 dpi. Genes in G7 were sharply upregulated in LS8 but mildly upregulated in SP at 5 dpi followed by down-regulation. Group G7 was enriched in the genes involved in the biosynthesis of secondary cell wall components, such as phenylpropanoid, cinnamic acid, and suberin. According to the reported etiology of VW infection, *V. dahliae* invaded plant tissues (usually roots) within 1 day, colonized the roots over the subsequent 3~5 days, and spreads over the vascular tissues of infected plants within 1~2 weeks [26,27]. Consistent with this infection process, resistant plants could initiate responses and reprogram gene expression and metabolism to modify cell walls, particularly those in vascular tissues, to prevent the expansion of VW infection.

In addition to G5 and G7, G13 comprises a small group of genes (120 genes) related to the biosynthesis of alkane, cuticle, wax, and suberin, with an expression trend of VW-induction in LS8 but low basal expressed in SP (Figure 5B). Moreover, genes in G12 and G14 shared a similar expression pattern in LS8 and downregulation in mock-treated samples but remained high expression in VW-inoculated LS8 samples. Both G12 and G14 groups were enriched in secondary cell wall-related functions, including xylan biosynthesis (GO:0045492), plant-type cell wall organization (GO:0009834), and hemicellulose metabolism (GO:0010410) (Figure 6). Notably, xylan is a predominant hemicellulose in grasses and has recently been recognized as critical for determining vascular tissue cell wall structures and biomass recalcitrance, mainly through β-1,4-linkages [28]. Thus, the enrichment of xylan biosynthetic genes that are upregulated in LS8 highlights xylan’s potential role in protecting vascular tissues from VW infection. Taken together, groups G5, G7, G12, G13, and G14 strongly support that, at 5 dpi, potato plants sense the propagation and spread of the VW pathogen along vascular tissue. In response, LS8 may upregulate the expression of defense/biotic-related genes, partly mediated by phytohormones, and inhibit VW pathogen propagation by enhancing the biosynthesis of secondary cell wall components, including hemicellulose, xylan, phenylprepanoids, lignin, cuticle, and suberin, to modify and strengthen cell wall integrity.

Many other genes (from G2 and G17) were induced by the mock treatment but downregulated in LS8; in contrast, these genes maintained low expression levels in SP, possibly representing LS8-specific responses to external stimuli. These genes were also enriched in GO terms including “response to external biotic stimulus”, “defense response”, and “regulation of jasmonic acid (JA)-mediated signaling”. Additionally, group G20 was rapidly induced in LS8 at 1 dpi and maintained high expression levels thereafter, but was barely expressed in SP (Figure 5B). G20 was characterized by defense-related terpenoid biosynthesis and the metabolism of salicylic acid (SA), a key phytohormone for pathogen resistance (Figure 6). This implies that the rapid perception of VW inoculation and subsequent SA- and specialized metabolite-mediated responses may contribute to VW resistance in LS8.

In contrast to the gene groups upregulated in LS8 but not in SP, groups G21, G22, G24, and G25 were upregulated in VW-inoculated SP samples, and the extent of upregulation was significantly higher than that in LS8 samples (Figure 5B). G21 was enriched in functions associated with catabolic processes, including the catabolism of organic substance, monosaccharides, and terpenoids (Figure 6). Similarly, G24 was associated with GO terms related to catabolism, such as “regulation of catalytic activity” (GO:0050790), “hydrogen peroxide catabolism” (GO:0042744). “Cellular oxidant detoxification” (GO:0098869), and “antibiotic metabolic process” (GO:0016999). G25 was linked to protein synthetic process, with enriched GO terms “ribosome” (GO:0005840) and “cytosolic large ribosomal subunit” (GO:0022625). The functional enrichment results of these SP-upregulated gene groups indicate that upon VW inoculation, SP plants perceive the pathogen, potentially leading to ROS and alterations in specialized metabolic pathways. Consistent with the leaf yellowing observed following long-term VW inoculation, leaf senescence was associated with protein degradation; in turn, inoculated cells in SP might transcriptionally enhance protein synthetic machinery to delay VW-associated cell death. Furthermore, G26 comprises genes that were more highly expressed in SP than in LS8, even though their expression was suppressed by VW inoculation. This group was enriched in functions such as “organ senescence”, “ABA signaling pathway”, and “superoxide anion generation”. These results suggest that abiotic stress responses might be more sensitive in SP than those in LS8, and that these pathways could be repressed on VW inoculation. In addition, G27 contains 20 genes whose expression was specifically upregulated in SP at 3 days post VW inoculation. G27 was enriched in genes involved in phenylpropanoid and lignin biosynthesis, indicating that specific components of the lignin biosynthetic pathway could respond to VW inoculation in SP.

Since our detailed interpretation of the functional enrichment results for different VW-associated gene groups pinpoints the involvement of G5, G7, G12, G13, and G14 in the metabolism of secondary cell wall components (e.g., hemicellulose, lignin, cinnamic acid, and suberin), we sought to further validate experimentally the association between cell wall metabolism with VW resistance in LS8. Phenylalanine ammonia-lyase (PAL) is the first enzyme in the phenylpropanoid pathway, and it modulates the metabolic flux of monolignol synthesis [29,30]. More importantly, PALs have been reported to confer resistance against multiple plant pathogens across various crops. In rice, *OsPAL4*, together with three other *OsPALs*, forms a *PAL* gene cluster that colocalizes with QTL for bacterial blight and sheath blight disease resistance [29,31,32]. More recently, studies have demonstrated that the rice transcription factor DWARF 53 (D53) directly represses the expression of *OsPAL1-7*, leading to reduced lignin accumulation and compromised resistance against *Magnaporthe oryzae* [32]. In soybean, GmPAL2.1 functions as a positive regulator of resistance against the pathogen *Phytophthora sojae* potentially by increasing the content of glyceollins, daidzein, genistein, and SA [33]. Heterologous expression of pepper (Capsicum annuum) CaPAL1 in transgenic Arabidopsis plants enhances resistance to infections by *Pseudomonas syringae* pv. tomato (*Pst*) and *Hyaloperonospora arabidopsidis* infection [34].

Given the critical role of PALs, we performed a BLASTp search (version 2.11.0; E value < 1 × 10^−10^) using the PAL proteins from *Arabidopsis thaliana* and sorghum as queries (Appendix A). Combined with protein domain validation (the aromatic amino acid lyase domain, PF00221), we identified 14 PAL-encoding genes in the potato C88 genome (Appendix A). These 14 *StPAL* genes are distributed across three homologous loci on chromosomes 3, 9, and 10, respectively. Considering that potato is a tetraploid species, *StPAL*s are present on all four homologous loci (C88_C10H1G019600, C88_C10H2G055500, C88_C10H3G078090, and C88_C10H4G106460 on chr 9, and C88_C09H1G006120, C88_C09H2G035670/C88_C09H2G035680, C88_C09H3G066480, and C88_C09H4G094610/C88_C09H4G094620 on chr 10). However, on the chr 3 locus, *StPAL*s were only detected on three homologous loci (C88_C03H1G004580, C88_C03H3G074930, and C88_C03H4G109900/C88_C03H4G109930). Notably, *StPAL*s have undergone expansion, likely driven by tandem duplication events, which resulted in the formation of gene pairs, including C88_C03H4G109900/C88_C03H4G109930, C88_C09H2G035670/C88 _C09H2G035680, and C88_C09H4G094610/C88_C09H4G094620 [28]. Since potato is a dicotyledonous species, we conducted a phylogenetic analysis of StPAL proteins with AtPALs and tomato (*Solanum lycopersicum*) SlPALs [30,35]. Our phylogenetic analysis revealed that these PAL proteins cluster into five groups (i.e., groups I, II, III, IV, and V; Figure 7A), with AtPAL1-AtPAL4 all grouping together with SlPAL (Solyc05T002733). Group-III PALs exclusively contain several SlPALs, while Groups I, IV, and V contain both SlPALs and StPALs (Figure 7A). Despite the expansion of *StPAL* genes, the encoded protein sequences show high similarity and contain highly conserved domains, suggesting that these StPALs are likely to be functional (Appendix A). Alignment of StPAL protein sequences and calculation of pairwise protein identities confirmed that all StPALs share high sequence similarity [36]. Specifically, StPALs derived from homologous loci exhibit higher similarity than those from different loci (Figure 7B). The high protein sequence similarity among StPALs and the strong conservation of the PAL domain indicate that all StPALs may be functional (Figure 7B and Appendix A; Appendix A).

Expression analysis revealed a distinct bias in *StPAL* expression towards loci on chr 9 and chr 10, whereas the *StPAL*s from chr 3 exhibited extremely low expression levels (Figure 7C). In contrast, several *StPAL*s located on chr 9 and chr 10 were highly expressed. The expression levels of these StPALs (from chr 9 and chr 10) were significantly upregulated upon mock and VW treatments in both SP and LS8, likely reflecting a natural wound repair response. Notably, LS8 samples consistently showed higher *StPAL* expression levels than SP samples under these treatments (Figure 7C). Since LS8 had higher StPAL expression levels than that in SP, we further measured PAL enzymatic activity in mock- and VW-inoculated samples in both cultivars. In both LS8 and SP, inoculation with VW enhanced PAL activity; however, LS8 exhibited more rapid PAL activity upregulation and maintained higher enzymatic activity levels compared to SP (Figure 7D). In LS8, PAL activity was elevated at 1 dpi, with significantly higher PAL activity consistently detected from 1 to 5 dpi (Figure 7D). This finding aligns with the VW resistance in LS8.

PALs have long been established as a key regulator of crop resistance against various pathogens and insects. Genetic analysis has identified *OsPAL4* and *OsPAL6* as colocalized with quantitative trait loci (QTL) for multiple rice diseases, including bacterial blight, sheath blight, and rice blast [29]. A recent study further demonstrated that *OsPAL6* and *OsPAL8* expression contributes to brown planthopper resistance by mediating salicylic acid and lignin biosynthesis [37]. Additional studies have highlighted the potential of coordinating multiple rice PALs to achieve broad-spectrum resistance against pathogens and insect pests [38]. Notably, the dynamic changes in PAL activity in either LS8 or SP did not fully correspond to *StPAL* gene expression profiles. This discrepancy underscores the importance of post-transcriptional regulation in modulating PAL activity during pathogen resistance. This direction deserves further investigations. Not surprisingly, a recent rice study identified a mechanism for PAL post-translational modification. The protein arginine methyltransferases 5 (PRMT5)-mediated protein methylation on OsPAL1 enhances PAL enzymatic activity, leading to increased SA accumulation and improved pathogen resistance [39]. The rapid induction of PAL activity at 1 dpi in LS8 thus warrants future investigation.

## 3. Materials and Methods

### 3.1. Plant Materials and Experiment Design

Two potato (*Solanum tuberosum* L.) cultivars, Shepody (SP) and LS8, with contrasting VW-resistant phenotypes were selected by the Institute of Economic Crops, Heilongjiang Academy of Agricultural Sciences (HAAS). Potato seedlings with a similar vegetative stage were pot-grown at the greenhouse of the Institute of Economic Crops, HAAS, during the 2024–2025 season for VW (*V. dahliae*) or mock inoculation and subsequent sampling and RNA-seq analyses using a randomized completely block design with three replicates. Seedlings of the two potato cultivars were inoculated with the pathogen or mock (as control), and the inoculated tissues were sampled at 0 day (before the inoculation), 1 day, 3 days, 5 days, and 13 days post inoculation, with at least three plants inoculated each time per treatment per replicate. Previously, the time-series inoculation process of *V. dahliae* to potato plants was analyzed with the GFP-expressing *V. dahliae* strain. The results demonstrated that *V. dahliae* spores germinated within 1 day, and the resultant hyphae successfully infected the root tissue within 7 days; subsequently, the hyphae began to spread to the above-ground parts of the plants via the vascular tissue [23,24]. Based on this information, we sought to characterize the gene expression responses of potato plants inoculated with *V. dahliae*, with a specific focus on elucidating the transcriptomic differences between cultivars with varying VW resistance within the first two weeks of inoculation. All the seedlings (from both cultivars SP and LS8) were grown to 4–6 leaves and used for the inoculation experiment. The inoculation was performed with the tear-bottom inoculation method in which each plant was carefully uprooted and a piece of the main root and underground stem tissue was scratched to remove the periderm for pathogen inoculation [24,40]. For mock inoculation, the sterilized Czapek-Dox medium without *V. dahliae* strain was used. Inoculation of both mock and the *V. dahliae* strain was performed at 2:00 to 3:00 pm, with the sampling conducted at the same hour to avoid potential effects of circadian rhythm on gene expression. At each sampling timepoint, the main root and its adjacent underground stem tissues were collected from each plant, and the samples from three plants were pooled as a replicate to minimize individual effects. The same sample was aliquoted and used for both RNA-seq and PAL enzymatic activity measurement. After sampling, the tissue was snap-frozen in liquid nitrogen and stored at −80 °C before the analyses.

### 3.2. Pathogen Infection Assay

We inoculated potato plants with *V. dahliae*. The strain had been collected from the field in Inner Mongolia by the potato disease resistance improvement research team at the Heilongjiang Provincial Key Lab of Potato Biology and Quality Improvement. According to Koch’s Postulates, the *V. dahliae* strain had moderate pathogenicity [40]. The *V. dahliae* strain was identified as Race 2 and MAT1-2-1 based on the reported method [41,42].

The *V. dahliae* strain was stored at −80 °C and was first reactivated by culturing on potato dextrose agar (PDA) plates. It was then transferred to sterilized Czapek-Dox medium and incubated on a shaker at 25 °C for 3–4 days. After filtering the culture through gauze, the spore concentration was adjusted to 2 × 10^5^ spores/mL. When the potato seedlings had grown to approximately 4–6 leaves, we selected healthy, pest- and disease-free potato seedlings as experimental materials. The inoculated potato plants were placed in a greenhouse under controlled conditions of 22 °C and 60% relative humidity.

### 3.3. RNA Extraction, Library Construction, and Sequencing

Total RNA was isolated using the TRIzol reagent (Catalog No. 15596026CN; Invitrogen, ThermoFisher Scientific Co., Mexico City, MO, USA). The integrity and purity of the RNA samples were assessed through agarose gel electrophoresis, NanoDrop 2000 spectrophotometry (Thermo Fisher Scientific Inc., Waltham, MA, USA), and Agilent 2100 Bio-analyzer (Agilent Technologies Inc., Santa Clara, CA, USA). The mRNA libraries were constructed following the standard protocols for the short-read RNA-seq sequencing and sequenced with 150 bp paired-end sequencing protocol by using the BGI DNBSEQ-T7 platform (MGI Tech Co., Ltd., Shenzhen, China). Cutadapt (https://cutadapt.readthedocs.io/en/stable/; accessed by 6 January 2025) [21] and FastQC (version 0.11.5; https://www.bioinformatics.babraham.ac.uk/projects/fastqc/; accessed by 6 January 2025) were employed for the quality control of the sequencing data and the evaluation of the QC results.

### 3.4. RNA-Seq Analyses

After quality control of the sequencing data, the clean reads were mapped to the tetraploid potato genome of the cultivar C88 with Hisat 2 (version 2.0.1-beta) [12]. Genic reads, including both uniquely mapped and multi-mapped reads, were extracted with featureCounts (version v 1.6.0) for the calculation of gene expression levels (using Fragments Per Kilobase of transcript per Million mapped fragments, FPKM) [43]. The genic reads were used for the identification of differentially expressed genes (DEGs) with edgeR with the following criteria of DEGs used: log2Fold Change ≥ 1 or≤ −1 and *q* value < 0.05 [44]. With the expressed genes identified (defined as at least five reads per gene per replicate sample and averaged FPKM value > 0.5), the principal component analysis (PCA) across all samples was performed using the prcomp function from the stats package in R v4.2.0 [45]. Hierarchical clustering was used to evaluate the correlation between replicate samples. Since there were nine samples (one control, four mock-inoculated, and four VW inoculated; replicates not included) within each cultivar, we applied analytical step suitable for time-series transcriptomic data to capture the VW-responsive gene clusters for each cultivar while omitting intrinsic transcriptomic differences between LS8 and SP. First, we identified the 14,310 and 21,739 DEGs in the LS8 and SP samples, respectively; second, these LS8- and SP-DEGs were subject to a noise-robust time-series gene clustering method Mfuzz, which led to 40 and 45 clusters in LS8 and SP, respectively. Mfuzz offers meticulous classification of time-series gene expression trends (e.g., more than 20 different clusters) [25,46]. The LS8 and SP gene clusters (representing a group of genes with highly similar expression patterns) were further merged into modules by calculating the Pearson correlation coefficient (PCC > 0.78) among the eigengene expression of each cluster. VW-responsive modules were identified. Then, these analyses allowed us to focus on those genes associated with VW inoculation (i.e., those belonging to the VW-responsive modules in either LS8 or SP), once combining the gene expression data in LS8 and SP. Among all the 207 module combinations, only 119 module combinations represent those associated with VW inoculation in LS8 or SP (a total of 12,797 DEGs). By calculating the PCC values among the representative expression patterns of the 119 module combinations, together with hierarchical clustering of these DEGs, these 12,797 DEGs were separated into 33 groups (namely, G1 to G33). Among the 33 groups, those that could possibly explain the VW resistance differences between LS8 and SP were visually identified based on expression patterns and subject to functional enrichment analysis with gene ontology (GO) annotations. GO enrichment analysis was performed for each group by calculating the hypergeometric test (*q* < 0.05) with the ClusterProfiler package as previously described [47].

### 3.5. Phenylalanine Ammonia-Lyase (PAL)—Encoding Gene Identification and Phylogenetic Analyses

To identify the genes encoding phenylalanine ammonia-lyase (PAL), PAL protein sequences from *Arabidopsis thaliana* and sorghum were used as the query for blastp (E value < 1 × 10^−10^) to search against all the proteins encoded by the potato C88 genome [30,38]. The identified potential StPAL proteins (only those encoded by the primary transcript per geneID) were validated with InterProScan for the full-length PAL domain (PF00221). StPAL proteins were aligned with MUSCLE [48], with the protein identity between each two of the StPALs determined using the online tool of the EMBL-EBI database (https://www.ebi.ac.uk/Tools/msa/, accessed on 6 January 2025) according to the previous report [49,50].

PAL protein sequences from *Arabidopsis thaliana* and tomato, along with the StPALs identified in the present study, were used for the phylogenetic analysis [30,36]. These sequences were first aligned using MUSCLE, and a maximum-likelihood phylogenetic tree was constructed with 500 bootstrap replicates in the MEGA-X software [48,51].

### 3.6. Analyses of PAL Activity

Phenylalanine ammonia-lyase (PAL) activity was measured with three replicates by using a commercially available assay kit (catalog No. AKAM012M; Beijing Boxbio Science & Technology Co., Beijing, China). The samples for PAL enzymatic activity analysis were aliquoted from the same sample for RNA-seq analysis. After sampling, the tissue sample was snap-frozen in liquid nitrogen and stored in a −80 °C freezer. Then, one aliquot of 0.2 g tissue was ground to powder at 4 °C with the Tissuelyser-24L for subsequent enzymatic activity measurement (Shanghai Jingxin Co., Shanghai, China). After tissue grinding, the crude enzyme solution was extracted by 10,000× *g* centrifugation for 10 min at 4 °C. Briefly, the PAL activity measurement catalyzes L-Phenylalanine to trans-cinnamic acid, which can be readily quantified at 290 nm with spectrophotometry. The assay was conducted in strict compliance with the manufacturer’s instruction of the kit. The reaction mixture was incubated at 30 °C for 30 min, and the absorption value of *trans*-cinnamic acid was measured at 290 nm. The absorption values reflecting PAL enzyme activities were determined with a U-5100 ultraviolet–visible spectrophotometer (HITACHI, Tokyo, Japan).

## 4. Conclusions

Verticillium wilt (VW)-resistant potato germplasm represents a critical resource for not only breeding towards VW resistance but also for understanding the underlying mechanism of VW resistance and the identification of key genes conferring to the resistance. In our work, we provide, as far as our knowledge, the first comprehensive analysis of a time-series RNA-seq dataset for the VW resistance potato cultivar. We found that, while VW-resistant LS8 and VW-susceptible SP exhibited a huge difference in their transcriptomes (reflected by a total of 38,036 DEGs between LS8 and SP), VW inoculation induced dramatic changes in transcriptomes within each cultivar regardless of the VW susceptibility, leading to 14,310 DEGs among the LS8 samples and 21,739 DEGs among the SP samples. With time-series RNA-seq analyses, we revealed the dynamic complexed transcriptomic changes that occurred in each of the two cultivars in about two weeks post inoculation. In particular, we noticed the rapid upregulating response of pathogen-related phytohormone salicylic acid (SA) and secondary metabolites (e.g., cadinene and sesquiterpenes) at 1 dpi in LS8 but not in SP. We highlighted several groups of genes (i.e., G1, G10, and G19) related to polysaccharide and choline metabolism that were elevated at 3 dpi in LS8. More importantly, genes of the G5, G7, and G13 were upregulated at 5 dpi and the genes function in responses to phytohormone ABA, biotic stress, and bacterium and the biosynthesis of several SCW components (e.g., cellulose, suberin, cuticle, and wax). LS8 also had consistently higher expression levels of genes associated with lignin, xylan, and hemicellulose synthesis (reflected in G12 and G14). These time-series LS8-specific transcriptomic changes and the corresponding interpretation gain new molecular insights into the VW resistance of LS8, indicating that the rapid responses associated with SA and subsequent metabolic reprograming (likely carbohydrate re-channeling to SCW, such as cuticle and suberin, and defense-related specialized metabolites, such as terpenes) contribute to the VW resistance in LS8. These results underline that secondary cell wall biosynthesis and its potential role in the modification of cell walls in vascular tissues could be valuable for preventing *V. dahliae* from colonizing and propagating along potato’s vascular tissues. Along with RNA-seq-based insights, we identified highly expressed VW-responsive StPAL genes and confirmed higher PAL activities in LS8 than those in SP. In particular, the high expression levels of several *StPAL*s (i.e., C88_C10H1G019600, C88_C10H3G078090, and C88_C10H4G106460), together with the associated high PAL activities in LS8 VW-inoculated samples, support these *StPAL*s as candidate genes to enhance the VW resistance. Future research works could focus on the validation of these StPALs’ functions in VW disease resistance, possibly by increasing *StPAL* expression levels through overexpression approaches. Overall, our results provide the first transcriptomic insights into VW resistance in potato cultivars and candidate genes associated with VW resistance, deserving further functional investigations in the future.

## Figures and Tables

**Figure 1 plants-15-00026-f001:**
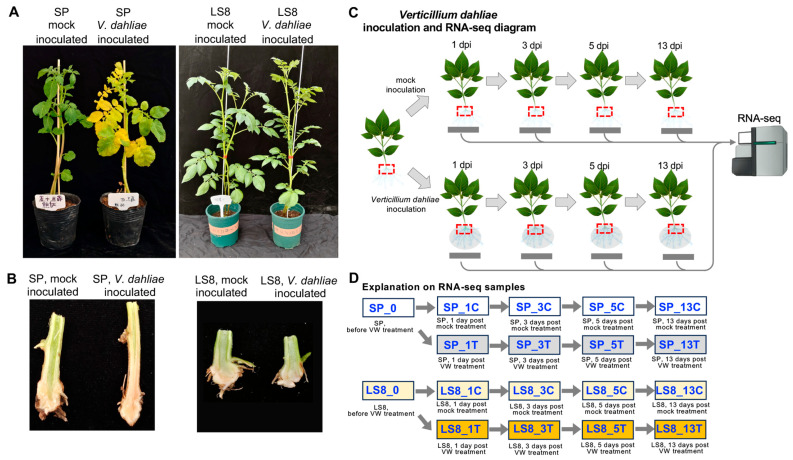
The Verticillium wilt (VW) inoculation experiment on two potato cultivars with contrasting VW resistance and the RNA-seq experiments. (**A**) After 30 days of VW inoculation, SP plants showed obvious yellow leaves compared to the control, while LS8 plants showed similar growth status without obvious difference in phenotypes compared with the control. (**B**) Detailed photos of the SP and LS8 inoculated underground stem tissues between the mock and VW inoculation experiments. After VW inoculation, the SP underground stem tissues began browning. (**C**) The diagram showing the VW inoculation experiment and the inoculated underground stem tissues (indicated with red boxes) were collected before inoculation and at 1 day, 3 days, 5 days, and 13 days post inoculation (dpi), with three replicates for each timepoint. Mock inoculation was used as the control. (**D**) Parallel experiments were conducted for both VW-susceptible and -resistant cultivars SP and LS8, respectively. The experiment was conducted in a greenhouse with three replicates, with each replicate including three similarly grown plants (one plant per pot). With this experimental design, for each cultivar, nine RNA-seq samples (including three replicates per sample) were collected for the RNA-seq analysis with the suffixes indicating the dpi and control or VW treatment (using “C” and “T”, respectively).

**Figure 2 plants-15-00026-f002:**
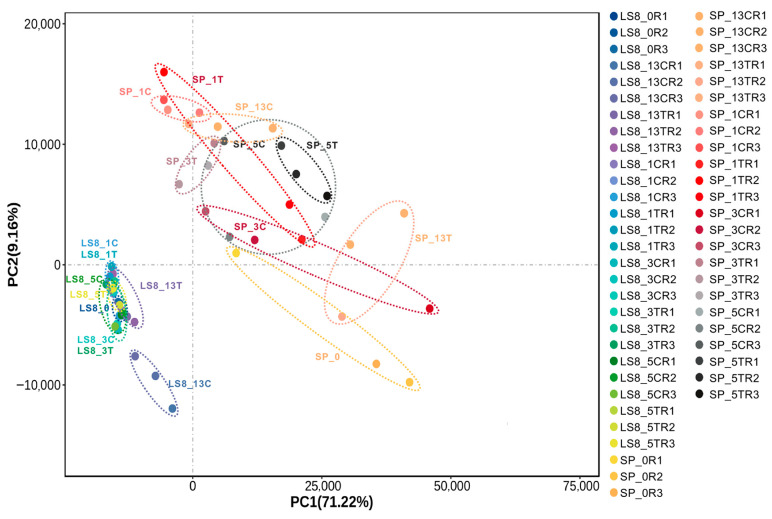
Principal component analysis (PCA) of the RNA-seq samples in this study. In the figure, each dot indicated an RNA-seq sample. The first principal component accounts for 71.2% difference between the samples, reflecting the differences between SP and LS8 cultivars and those between the control and treatment. From the PCA result, LS8 transcriptomes appeared to be much more staple after the VW treatment and could be barely distinguishable between the control samples and VW-treated samples. On the contrary, SP transcriptomes became dramatically changed, with the control and VW-treated samples distinguishable between each other.

**Figure 3 plants-15-00026-f003:**
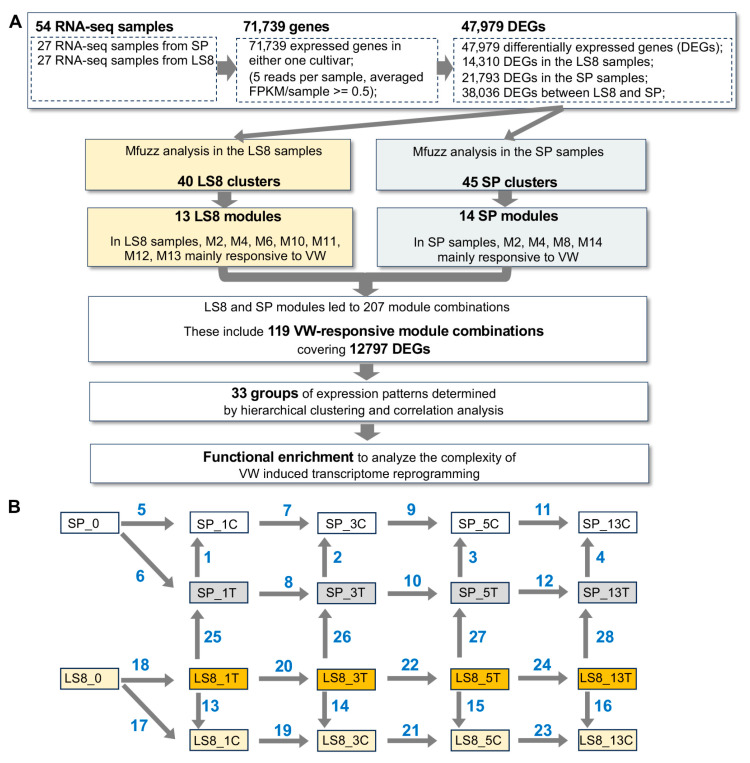
Differential expression analysis results and the bioinformatic analysis workflow to identify the VW-induced transcriptome changes. (**A**) To effectively capture the VW-induced transcriptomic changes and distinguish these VW-relevant changes from the intrinsic SP-LS8 gene expression differences, we built a comprehensive bioinformatic analysis workflow. Genes with an average FPKM ≥ 0.5 and more than 5 reads per replicate at one treatment per stage were considered as expressed. (**B**) A series of pairwise comparisons between RNA-seq samples were performed to identify the differentially expressed genes (DEGs; q-value < 0.05 and log2FoldChange ≥ 1 or ≤ −1). Blue number indicates the comparison groups. During the 14-day development, both the non-treated SP and LS8 plants experienced significant changes in their transcriptomes. For the VW treatment, both SP and LS8 samples also experienced transcriptome reprogramming, involving thousands of DEGs. Notably, more than 25,000 genes were differentially expressed between SP and LS8 VW-treated samples, highlighting the intrinsic transcriptomic differences between the two cultivars.

**Figure 4 plants-15-00026-f004:**
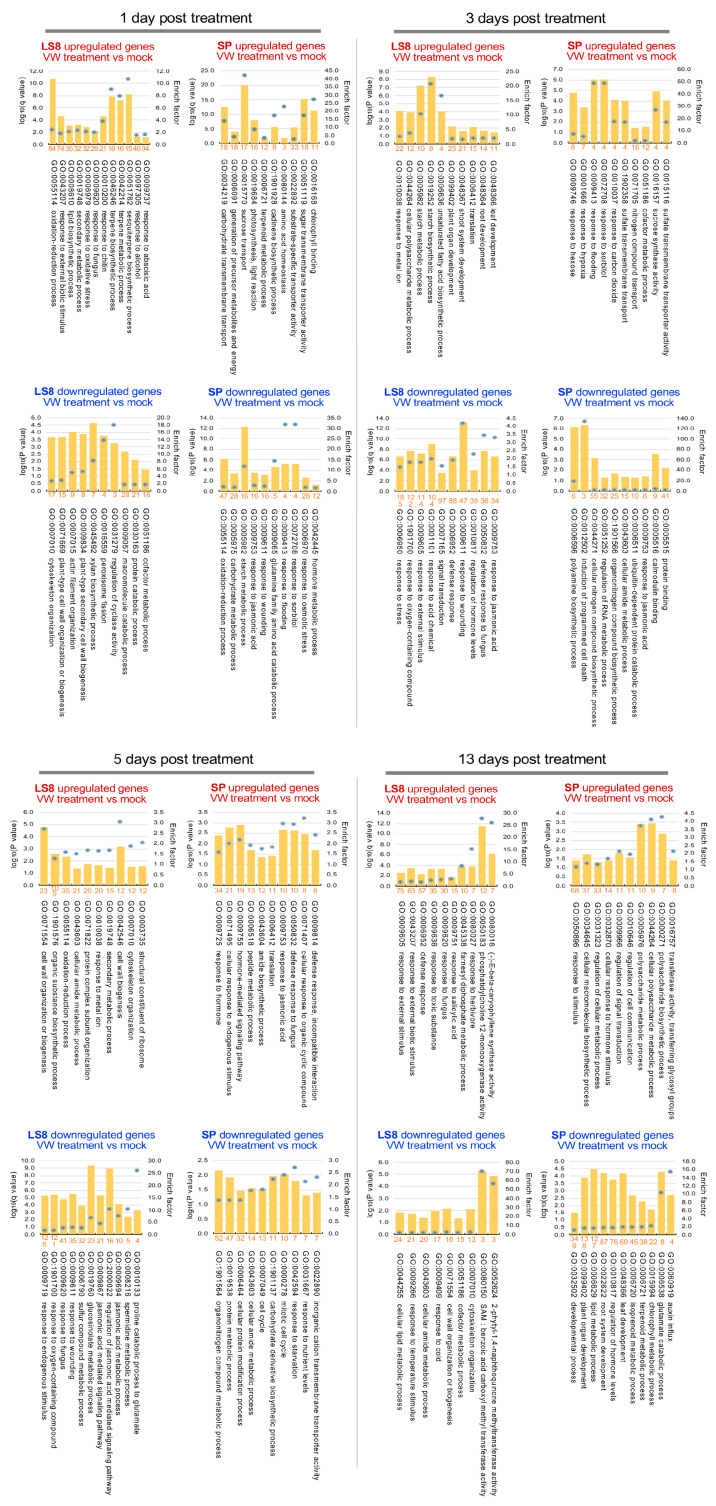
Representative gene ontology (GO) functional terms enriched in the LS8-upregulated, LS8-downregulated, SP-upregulated, and SP-downregulated genes at 1 dpi, 3 dpi, 5 dpi, and 13 dpi. Dpi, day post inoculation. Upregulated and downregulated genes are indicated with red and blue fonts. Both log_10_ (*p* value) or log_10_ (*q* value) and enrich factor for the GO terms are provided. The number of genes per enriched GO term are given in the orange font.

**Figure 5 plants-15-00026-f005:**
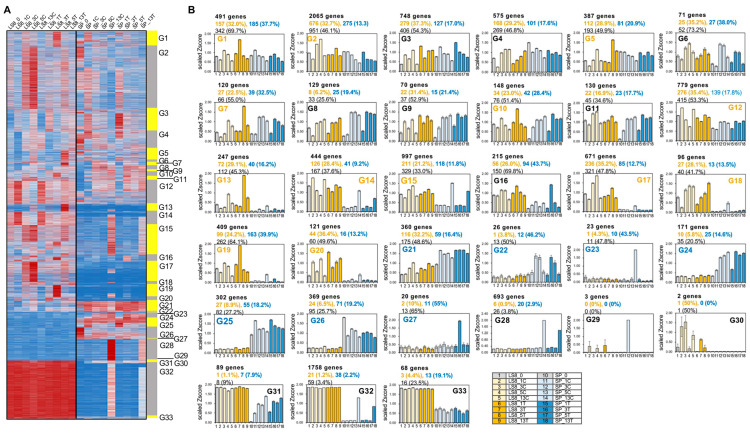
The 12,797 VW-responsive DEGs were separated into 33 groups of expression trends (G1 to G33). (**A**) The heatmap showing the expression patterns of the 12,797 DEGs from G1 to G33 using scaled Zscore, with the yellow and gray bars indicating different expression groups. The magnitude of red and blue colors indicates high and low expression. (**B**) The representative expression trends of each group are shown in bar plots, with the number of genes for each group provided. Data are presented as mean and standard error of the mean (S.E.M.). Among these groups, G1, G2, G5, G7, G10, G12, G13, G14, G15, G17, G18, G19, and G20 (group numbers shown in yellow) showed differential expression patterns between LS8 control stages and LS8 VW-treated stages, while G21, G22, G23, G24, G25, G26, and G27 (group numbers shown in blue) exhibited VW-responsive expression patterns in SP samples or dramatic differences between SP and LS8 samples. LS8 VW-responsive DEGs and SP VW-responsive DEGs are the DEG groups not related to wound responses common to the mock and VW treatments, representing valuable gene sets for screening the gene groups associated with VW resistance. The number of overlapped genes and percentages for each group with those LS8 VW-responsive DEGs (shown in yellow font), SP VW-responsive DEGs (shown in blue font), and LS8/SP combined VW-responsive DEGs (shown in black font) were calculated and are provide above each subfigure for the gene group expression patterns.

**Figure 6 plants-15-00026-f006:**
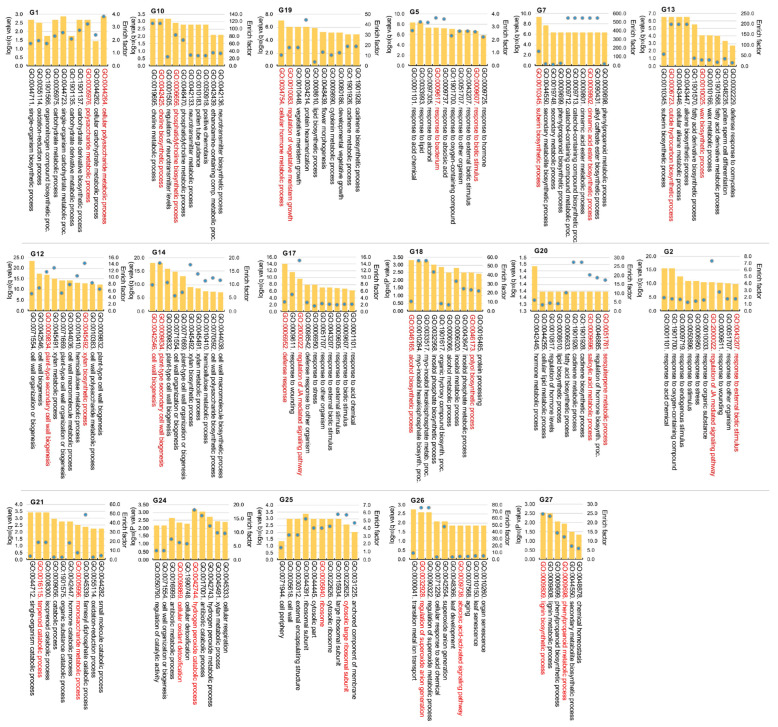
Representative gene ontology (GO) functional terms of each VW-responsive expression group. For each expression group, the top 10 biological process GO terms are visualized with the enrichment values and enrich factors provided (detailed methods in “Section 3.4”; Data S1). The significant enrichment values were determined by Hypergeometric test, *Q_hypergeometric_* < 0.05; for some groups, when the Q values for all the identified GO terms were more than 0.05, the *P_hypergeometric_* < 0.05 was used as the criteria. The most representative two GO terms for each group are highlighted in red. Enrich factors for the GO terms are shown in the secondary *y*-axis.

**Figure 7 plants-15-00026-f007:**
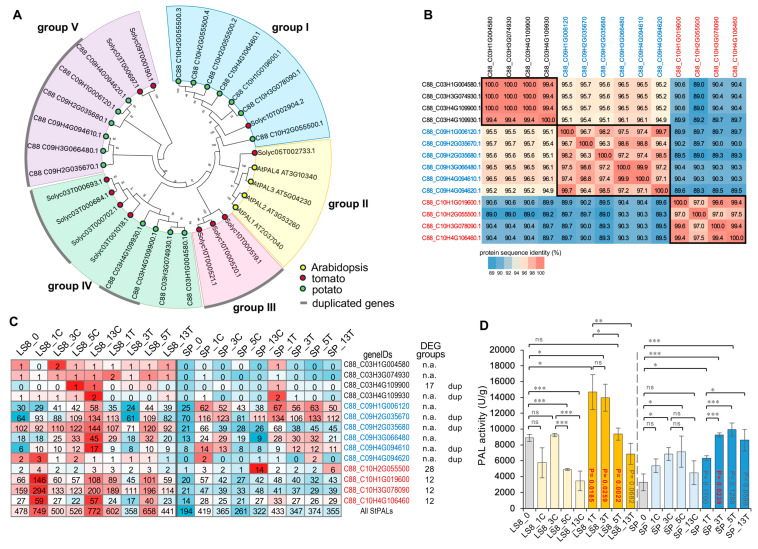
The high expression levels of phenylalanine ammonia-lyase (PAL) and PAL enzymatic activities in LS8 tissues were associated with the resistance to VW. (**A**) The phylogenetic tree consisting of the PAL proteins from Arabidopsis, tomato, and potato, were constructed with the maximum likelihood method (500 times bootstrap) by using MEGAX, with the supporting values for each branch provided. Protein sequences were aligned with MUSCLE before phylogenetic analysis. Gray arcs indicate the duplicated *PAL* genes in tomato and potato. (**B**) Protein identity values between the StPALs were calculated, highlighting high levels of sequence similarity. The blue and red colors indicate the protein sequence identity values. (**C**) The expression patterns of 14 *StPAL* genes between LS8 and SP samples. Genome-wide identification found 14 PAL-encoding genes and *StPAL*s were likely expanded due to gene duplication (indicated in the right panel of (**C**)). The blue and red colors in the heatmap reflect the magnitude of expression (in Zscore) among the samples, with the actual expression values (in FPKM) provided on the heatmap. Five *StPAL*s, belonging to G12, G17, and G28, respectively, are indicated. (**D**) PAL enzymatic activities were measured in LS8 and SP samples. Statistical differences in PAL activity between samples (either within LS8 or SP samples) were calculated using Student’s *t*-test, with significance denoted by asterisks (*, *p* < 0.05; **, *p* < 0.01; ***, *p* < 0.005; ns, not significant). *p* values comparing PAL activity between VW-treated and corresponding mock samples (for example, LS8_13T vs. LS8_13C) are provided in the bar plot, with significant *p* values highlighted in red font.

**Table 1 plants-15-00026-t001:** The comparison between replicated RNA-seq samples and the number of differentially expressed genes (DEGs) identified.

Comparison Group	Sample 1	Sample 2	Total No. of DEGs	No. of Upregulated DEGs	No. of Downregulated DEGs
1	SP_1T *vs*	SP_1C	1060	461	599
2	SP_3T *vs*	SP_3C	751	371	380
3	SP_5T *vs*	SP_5C	663	374	289
4	SP_13T *vs*	SP_13C	3075	327	2748
5	SP_1C *vs*	SP_0	3718	2130	1588
6	SP_1T *vs*	SP_0	2311	1456	855
7	SP_3C *vs*	SP_1C	1855	1019	836
8	SP_3T *vs*	SP_1T	996	522	474
9	SP_5C *vs*	SP_3C	97	50	47
10	SP_5T *vs*	SP_3T	318	193	125
11	SP_13C *vs*	SP_5C	6948	6572	376
12	SP_13T *vs*	SP_5T	299	164	135
13	LS8_1T *vs*	LS8_1C	1419	844	575
14	LS8_3T *vs*	LS8_3C	1351	389	962
15	LS8_5T *vs*	LS8_5C	1654	516	1138
16	LS8_13T *vs*	LS8_13C	1315	719	596
17	LS8_1C *vs*	LS8_0	1107	564	543
18	LS8_1T *vs*	LS8_0	1195	769	426
19	LS8_3C *vs*	LS8_1C	1453	1089	364
20	LS8_3T *vs*	LS8_1T	993	474	519
21	LS8_5C *vs*	LS8_3C	147	67	80
22	LS8_5T *vs*	LS8_3T	2005	1235	770
23	LS8_13C *vs*	LS8_5C	3726	1590	2136
24	LS8_13T *vs*	LS8_5T	1788	1012	776
25	LS8_1T *vs*	SP_1T	25,799	13,378	12,421
26	LS8_3T *vs*	SP_3T	26,031	13,035	12,996
27	LS8_5T *vs*	SP_5T	27,072	14,084	12,988
28	LS8_13T *vs*	SP_13T	12,986	5546	7440

## Data Availability

The data presented in this study are available in the article and the Appendix A.

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
