# Peer review of "Time-Series Comparative Transcriptome Analyses of Two Potato Cultivars with Different Verticillium Wilt Resistance"

_plants, 2025, doi:10.3390/plants15010026_

Round 1

Reviewer 1 Report

Comments and Suggestions for Authors

Fan et al. used a transcriptomic approach to investigate time-course expression differences between a susceptible (SP) and a resistant (LS8) potato cultivar to Verticillium wilt (VW).  The authors found different expression profiles between the cultivars and when inoculated by VW, and identified groups of genes that showed different expression profiles at different time points tested.  This manuscript is of general interest and presents new findings on the response of these cultivars to VW.  There are some important aspects of the manuscript that need to be addressed, though. 

  1. The experimental design and materials and methods are not clear and more details are needed for this project to be repeated/fully evaluated. See the PDF with specific comments.
  2. It is unclear what samples were compared when determining DEGs. This is central to data interpretation, but line 182 just says they used “pair-wise comparisons between different samples”.  What were these samples?  It should be the mock vs treatment inoculated samples at each time point, but Figure 3 suggests a with-in treatment comparison.  This needs to be very clear and the figure should be accurately updated, as it is the basis for all downstream analyses and discussions.    
  3. Similarly, Figure 4 displays expression for all samples at all timepoints. It makes more sense to display this data as differential expression between the mock and treatment inoculate samples per timepoint.
  4. The manuscript is also unclear in the “merging of the gene expression matrix from 13 LS8 and 14 SP modules” (Lines 215). Was this just a way to narrow down the DEGs evaluated, i.e. only including DEGs from each cultivar?  More details on this approach is needed, since the way it is written suggests they were selected and grouped was “based on the similarity of expression profiles” (Line 221). 
  5. I would be interested in seeing a GO or pathway analysis for the DEGs in the LS8 mock vs treatment compared to SP mock vs treatment (either pooled or per time period). An interaction analysis could also identify genes significantly differentially expressed between LS8 and SP. This may more directly address the question of genes and pathways involved on resistance in LS8.   
  6. The PAL enzyme activity analysis needs more detail about how and when these enzymes were isolated, especially when connecting activity to the expression profiles.
  7. Please see the attached PDF for additional comments and edit recommendations.

Reviewer 2 Report

Comments and Suggestions for Authors Title: It'd be modified like: 'Time-Series Comparative Transcriptome Analyses of two Potato Cultivars with Different/Contrasting Verticillium wilt Resistance' Abstract: pls. fix an issue with comma in line 26. Lines 36-37: pls. introduce the 'PAL' abbreviation.  Intro: I'm wondering why the similar findings (eg 10.3390/plants14091281) on potato VW have not been mentioned in this part?..  MM: 3.1 Why were these five DPIs selected for RNA-seq? Are there any bases for selecting this time series? 3.2 What specific strain of V. dahlie was used for the inoculation? At least, culture collection identifier needs to be indicated. If available, the strain virulence characteristics should be highlighted as well. Line 457 - company manufacturing the platform should be included. It's always good to run a phylogenetic analysis of the studied (PAL) genes with the same ones annotated from different species to show their homology (heatmap 6b doesn't  show it).     Results and discussion: it's the right choice to merge these two sections. However, the discussion part is actually minimized. It means that one can mainly see the analysis findings obtained. It's strongly suggested to find the previous studies (even for other hosts/genes with similar function) and compare them to yours. Also, some foreword before each subsection can help with the understanding of the studied subject (as part of discussion). The section seems to be slightly  overloaded with figures: such figures as workflow or expression trends or GO functional terms can be moved to supplementary data. More info on PAL  genes in terms of plant defense can be provided. Since it's not shown in the intro, it can be highlighted somewhere in the lines 375-80.     Conclusion: it's suggested to point out which gene engineering (eg RNA silencing) can employ the obtained results for VW control and management.  --- also, some very minor English language related issues can be fixed when proofreading the text 

Round 2

Reviewer 1 Report

Comments and Suggestions for Authors

I would like to thank the authors for their thorough edits and responses to my previous comments.  The additions and edits they have made to the materials and methods and to the text have greatly improved the clarity of the manuscript and project.

The addition of the mock vs VW inoculation analysis at each timepoint helped provide some context for the genes involved in a response to VW.  I like the addition of Table 1 and Figure 4 to display these results.  To clarify, I agree with the authors that the time course analysis is a strong approach to identify genes involved in a response to VW.  The benefit of the mock vs VW analysis is to identify genes that are likely involved in a response to the VW and not just to the wounding that occurred during inoculation.  Since both mock and VW inoculation entailed scratching the periderm, these wound-repair genes are likely expressed in both the mock and VW inoculated samples.  Therefore, the DEGs detected between the mock and VW inoculation sample are likely related to the VW infection response and not just wound healing, but the authors are correct the time course provides extra insight into the response and might catch genes not detected as differentially expressed at a specific timepoint (but comparisons to 0 dpi would also help).  This is of specific interest in parsing out the role of lignin biosynthesis and cell wall biosynthesis genes in VW resistance, since these gene families are likely involved in both wound repair and resistance.  I would suggest adding a sentence or two specifying how your approach differentiates between the genes involved in a response to wounding vs response to VW.    

To further improve the manuscript, I had additional minor suggestions in the attached PDF.  Specifically, Figure Legend 3 is long and cites other figures. It might be better to shorten the figure legend and include these details as a paragraph in the results instead of in the figure legend. I made other minor language suggestions.  I would suggest proofreading again to catch other grammar mistakes. 

Author Response

I would like to thank the authors for their thorough edits and responses to my previous comments. The additions and edits they have made to the materials and methods and to the text have greatly improved the clarity of the manuscript and project.

The addition of the mock vs VW inoculation analysis at each timepoint helped provide some context for the genes involved in a response to VW. I like the addition of Table 1 and Figure 4 to display these results. To clarify, I agree with the authors that the time course analysis is a strong approach to identify genes involved in a response to VW. The benefit of the mock vs VW analysis is to identify genes that are likely involved in a response to the VW and not just to the wounding that occurred during inoculation. Since both mock and VW inoculation entailed scratching the periderm, these wound-repair genes are likely expressed in both the mock and VW inoculated samples. Therefore, the DEGs detected between the mock and VW inoculation sample are likely related to the VW infection response and not just wound healing, but the authors are correct the time course provides extra insight into the response and might catch genes not detected as differentially expressed at a specific timepoint (but comparisons to 0 dpi would also help). This is of specific interest in parsing out the role of lignin biosynthesis and cell wall biosynthesis genes in VW resistance, since these gene families are likely involved in both wound repair and resistance. I would suggest adding a sentence or two specifying how your approach differentiates between the genes involved in a response to wounding vs response to VW.

To further improve the manuscript, I had additional minor suggestions in the attached PDF. Specifically, Figure Legend 3 is long and cites other figures. It might be better to shorten the figure legend and include these details as a paragraph in the results instead of in the figure legend. I made other minor language suggestions. I would suggest proofreading again to catch other grammar mistakes.

Answer: Thank you very much for your constructive comments and suggestions. By communicating with you through this peer-review process, we have actually evolved to reach better analyses for this RNA-seq data set.

We completely agree with you on that comparison between VW-treated and mock-treated samples in either LS8 or SP can identify the DEGs that are VW responsive and avoid those associated with the common wound response and/or repair processes. Consistent with this notion and your 2nd round comments, the two RNA-seq analytical approaches should joint at a certain stage to further narrow down candidate gene sets.

In agreement with your comments, we calculated the LS VW-responsive DEGs and SP VW-responsive DEGs by merging the VW-responsive DEGs across the four days post inoculation for each cultivar (provided in Table S4). This benefit has been added and clearly mentioned in the results (Lines #265-268).

While our time course analytical approach has its advantage, still it can leverage the LS VW-responsive DEGs and SP VW-responsive DEGs to highlight, among 33 gene groups in Figure 5,the gene groups in which “true” VW resistance-associated genes may be enriched. To address this, we compared the 33 gene groups with LS VW-responsive DEGs, SP VW-responsive DEGs, and their combination (namely LS8/SP VW-responsive DEGs). The number of overlapped genes between each gene group with LS VW-responsive DEGs, SP VW-responsive DEGs, and LS8/SP VW-responsive DEGs were determined and are labeled in the revised Figure 5. After this analysis, we clearly found that in several gene groups, LS8 VW-responsive DEGs and SP VW-responsive DEGs comprise high percentages (more than 45% of the total number in the corresponding gene set), suggesting that these gene groups should enrich the DEGs identified by both the aforementioned two RNA-seq analytical approaches. These gene groups include G1, G2, G5, G7, G10, G12, G13, G17, G18, G19, G20, G21, G22, and G23. This indicates that the functional enrichment and corresponding interpretations for these gene groups should be more relevant to the phenotypic difference in VW resistance between the two cultivars.

For the aforementioned analysis and corresponding revision, we have now revised Figure 5, Table S4, and the main text (Lines #265-268; Lines #418-424; Lines #425-435). Finally, we are grateful for your efforts and communication during this peer-review.

Regarding the Figure 3 legend.Thank you for this constructive comment. We have revised and shortened the Figure 3 legend (Lines #248-258). The revised legend od Figure 3 now has 147 words. The issue of citing the figure itself and other figures has been corrected.
Many sentences in the original legend have been moved to the Results section and merged together with result description (Lines #354-370).

Responses to the reviewer’s detailed comments in the manuscript PDF:

Answer:
Line #53. “up-ground” has been changed to “above-ground”.
Line #56. Regarding the repetition of “causing economic losses of 10% to 53%”, this error has been revised and removed.
Line #142. We have revised this sentence to mention the tissue used for RNA-seq sampling. (now at Lines #141-143)
Line #150. A new paragraph starts now. (Line #151)
Lines #164-167. This sentence has been revised accordingly. (Lines #164-166)
Figure 1C. The “explanation” word has been capitalized. (Line #168, Figure 1C)

Line #179. “included” has been revised to “including”.
Line #190. “accounts” has been revised to “account”.
Line #221. “stages” has been revised to “days post inoculation (dpi)”.
Line #224. “hold” has been changed to “holds”.
Line #247. In the Figure 3 title, “Different” has been changed to “Differential”.
Lines #247-258. Regarding Figure 3 legend, several revisions have been made: (1) We now do not cite any figures and tables in the figure legend. (2) The figure legend has been shortened to ~150 words. Many sentences have been removed to the Results section (Lines #354-370). (3) “gene The ontology” typo has been corrected (and now been removed to the Results section).
Line #262. “DPI” has been revised to “dpi”. This error has been checked throughout the manuscript.
Line #314. “remain” has been revised to “continued”.
Line #319. “suggestive” has been revised to “suggesting”.
Line #319. “important” has been revised to “are important”.
Line #365 (now at Line #345). “differed” has been changed to “differences”.
Line #431 (now at Line #411, 413 of the Figure 5 legend). These color-coded group numbers have been explained in the revised figure legend.
Line #454 (now at Line #437). “V. dahliae” has been changed to “V. dahliae”.
Line #539 (now at Line #525). The sentence has been revised as “Considering that potato is a tetraploid species”.
Line #614 (now at Line #602). “pot grown” has been revised to “pot-grown”.
Line #620 (now at Line #608-612). The sentence has been revised as follow.Previously, the time-series inoculation process of V. dahliae to potato plants was analyzed with the GFP-expressing V. dahliae strain. The results demonstrated that V. dahliae spores germinated within 1 day and the resultant hyphae successfully infected the root tissue within 7 days; subsequently, the hyphae began to spread to the above-ground parts of the plants via the vascular tissue.
Line #623 (now at Line #612). “up-ground” has been changed to “above-ground”.
Line #624 (now at Line #613-615). The sentence has been revised as follow.
Based on this information, we sought to characterized the gene expression responses of potato plants inoculated with V. dahliae, with a specific focus on elucidating the transcriptomic differences between cultivars with varying VW resistance within the first two weeks of inoculation.Line #629 (now at Line #618). “with” has been changed to “and”.
Line #635 (now at Line #623-626). The corresponding sentence has been revised to describe the detailed sampling tissue.
Line #643 (now at Line #633). “Koch’s postulates” has been changed to “Koch’s Postulates”.
Line #651 (now at Line #640-641). The temperature and relative humidity in the greenhouse have been specified.
Line #709-713 (now at Line #699-702). The corresponding method description (about phylogenetic tree construction) has been revised to remove grammatical errors.
Line #719 (now at Line #708). “refrigerator” has been changed to “freezer”.
